# Killing a zombie: a full deletion of the BUB1 gene in HAP1 cells

Jonne A Raaijmakers & René H Medema

The EMBO Journal (2019) 38: e102423

Comment on: **G Zhang** *et al* (April 2019)

See reply: **G Zhang** *et al* (November 2019)

The exact contribution of the BUB1 kinase to the spindle assembly checkpoint (SAC) in mammalian cells has been under debate since many years. While some studies confirmed a (near)-essential role for BUB1 in the SAC (Meraldi & Sorger, 2005; Klebig *et al*, 2009), other studies reported no obvious SAC defect in Bub1-deficient cells (Johnson, 2004) or showed a SAC defect that could only be observed after sensitization with inhibitors for Mps1 (Vleugel *et al*, 2015). The arrival of CRISPR technology has thus far also failed to resolve this controversy. We and others showed that deletion of BUB1 in HAP1 or RPE-1 cells resulted in only a minor SAC defect (Currie *et al*, 2018; Raaijmakers *et al*, 2018), while another study showed that (acute) deletion of BUB1 in p53-deficient RPE-1 cells results in a more prominent SAC defect (Rodriguez-Rodriguez *et al*, 2018). The latter study showed that editing of the BUB1 locus by CRIPSR-Cas9 is challenging as alternative splice variants of BUB1 can be expressed. Recently, Zhang *et al* combined CRISPR/Cas9-mediated gene editing with siRNA depletion and concluded that BUB1 also makes a prominent contribution to the SAC in HeLa cells (Zhang *et al*, 2019). In addition, this study suggested that the role of BUB1 in the SAC was overlooked in HAP1 and RPE-1 cells due to residual expression of BUB1 protein (2 and 8%, respectively), and subsequently showed that the SAC defect became more apparent when these cells were further challenged with BUB1 siRNAs. Thus, complete deletion of BUB1 can be challenging. A recent commentary therefore referred to BUB1 as a zombie protein, and it was suggested that the only way to completely delete BUB1 is to remove the entire gene (Meraldi, 2019).

Here, we did exactly that. We deleted the BUB1 gene in HAP1 cells using CRIPSR/Cas9 by using two-guide RNAs targeting the first and last exon of BUB1 (Fig 1A, Appendix Supplementary Methods, Appendix Table S1). Clones that displayed successful loss of the two targeted exons were tested for the absence of several additional exons located in the gene body. We identified two ΔBUB1 clones that met all criteria; none of the tested exons were present in clones 20 and 35 (Fig 1B). This implies that in these clones, the entire BUB1 gene was deleted successfully and the deleted fragment was not integrated somewhere else in the genome. Clone 20 displayed a fusion between exons 1 and 24, involving the break sites induced by Cas9 (Fig 1B and C). In clone 35, we were unable to amplify such repair product (Fig 1B), and therefore, we are uncertain of how the genomic locus in clone 35 was precisely repaired. Nonetheless, it is clear from the loss of all exons that also in this clone, the BUB1 gene locus is deleted in its entirety. As expected, both clones did not display any BUB1 transcripts (Fig 1D). Besides, no BUB1 could be observed by Western blot or immunofluorescence (Fig 1E, H and I). Also, H2A-pThr210, a well-characterized substrate of the BUB1 kinase, was completely absent in the two ΔBUB1 clones (Fig 1F and G). Furthermore, and consistent with our previous observations, BUBR1 levels at kinetochores were severely reduced in the ΔBUB1 clones, although some residual levels could still be observed (Fig 1G and J, Raaijmakers *et al*, 2018). Taken together, we successfully

generated two ΔBUB1 clones in HAP1 cells by completely eradicating the entire gene locus.

Next, we assessed the SAC functionality in the full knockout clones (ΔBUB1 c20 and c35). To this end, we performed live cell imaging of cells treated with a high dose of nocodazole to trigger a full SAC response. We observed that both ΔBUB1 clones were able to establish a functional SAC as cells displayed a prominent arrest in response to nocodazole treatment, not significantly different from WT HAP1 cells or our previously published BUB1 KO cells (ΔBUB1 "Ex3"), that were generated by the stable integration of a Blasticidin resistance cassette in exon 3 (Fig 2A, Raaijmakers *et al*, 2018). Depleting BUB1 with an siRNA did not affect the SAC response in any of the tested clones (Fig 2A). However, we observed previously that our ΔBUB1 "Ex3" cells failed to maintain a prominent SAC when treated with a low dose of the MPS1 inhibitor reversine. Consistently, the ΔBUB1 clones c20 and c35 were unable to maintain a SAC when challenged with a high dose of nocodazole in the presence of a low-dose reversine, while WT HAP1 were still able to arrest (Fig 2B). These data suggest that BUB1 becomes critical to maintain a SAC when the SAC is not fully activated. To test this, we induced a more graded SAC response by treating the cells either with a lower dose of nocodazole or with noscapine, an opium alkaloid that produces minor perturbations in chromosome alignment, to the extent that only one or a few chromosomes misalign (Tame *et al*, 2016). Consistent with this hypothesis, we found that the full BUB1 knockout, ΔBUB1 c20, was unable to maintain a SAC under conditions where the SAC was

Division of Cell Biology, Oncode Institute, The Netherlands Cancer Institute, Amsterdam, The Netherlands. E-mails: j.raaijmakers@nki.nl; r.medema@nki.nl
**DOI** 10.15252/embj.2019102423 | The EMBO Journal (2019) 38: e102423 | Published online 22 October 2019

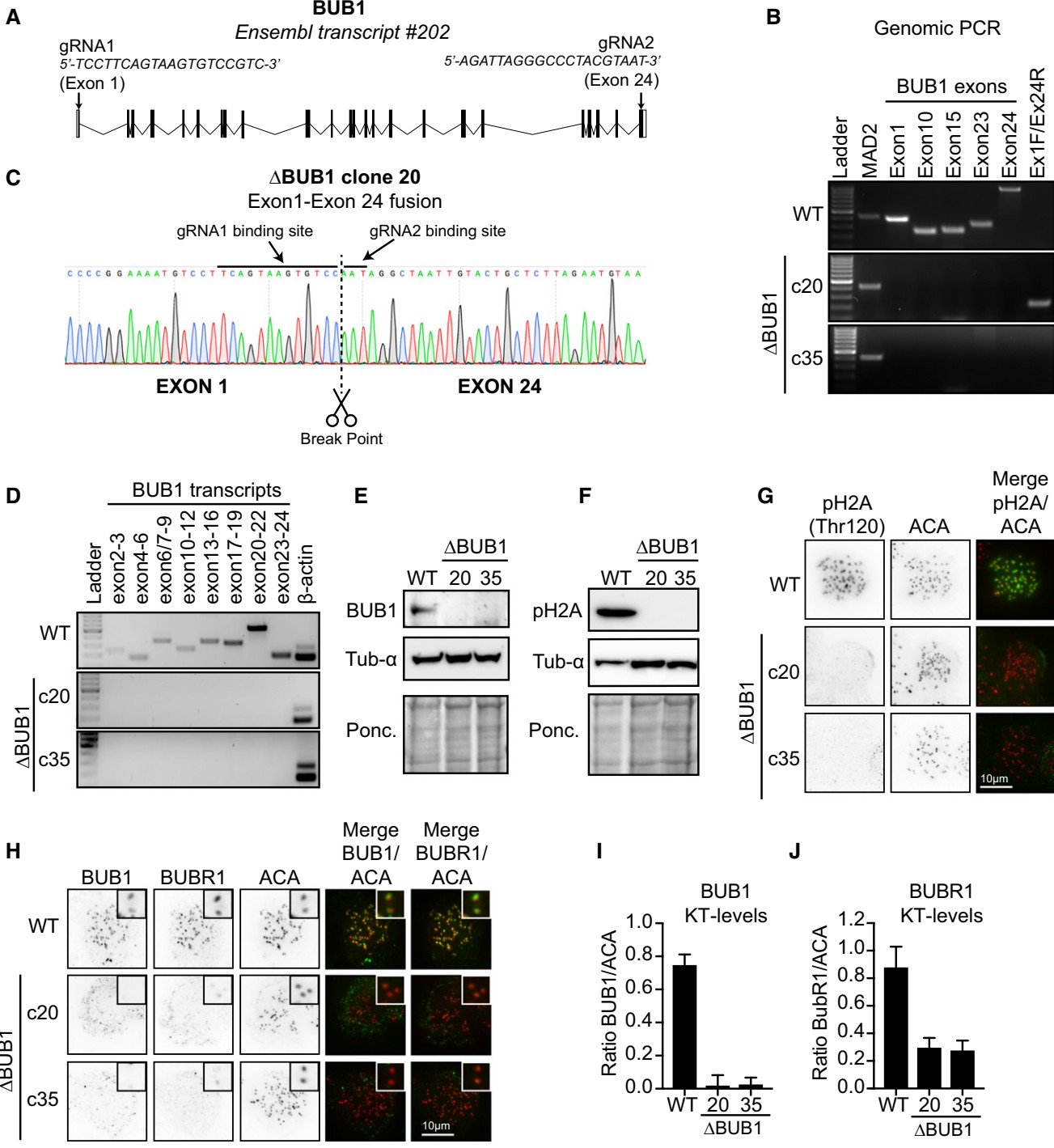

**Figure 1. Generation and characterization of full BUB1 gene deletion cell lines.**

(A) Schematic overview of the genomic BUB1 locus and targeting gRNAs used to generate the BUB1 knockouts. (B) Whole-genomic DNA was extracted from WT HAP1 cells and from each indicated clone and analyzed by PCR for the presence of the indicated BUB1 exons. A product amplifying the first exon of MAD2 was used as a positive control. Primers spanning exon 1 and exon 24 amplified a product in clone 20. (C) The product amplified form clone 20 using primers for exon 1 and exon 24 was analyzed by Sanger sequencing. A fusion between exons 1 and 24 could be confirmed and was induced at the guide binding sites. (D) BUB1 transcripts were amplified from cDNA generated from WT HAP1 cells or the two ΔBUB1 clones. Amplification of β-actin was used as a positive control. Primer pairs were designed as such that they at least spanned 1 intron to ensure specific amplification. (E, F) WT and ΔBUB1 clones were analyzed by Western blot. α-Tubulin and Ponceau S were used as loading controls. (G, H) Immunofluorescence images of WT and ΔBUB1 cells treated with nocodazole for 3 h before fixation. Cells were stained for either H2A-pThr210 or BUB1 (green) combined with BUBR1 (green) and centromeres/ACA (red). Scale bar: 10 μm. (I, J) Quantification of kinetochore signals in (H). All visible kinetochore pairs in at least 10 cells were quantified, corrected for cytoplasmic levels, and normalized for ACA.

Data information: See Appendix Supplementary Methods for experimental details.

graded (Fig 2C and D). In contrast, we found that our previously generated ∆BUB1 clone "Ex3" was in fact able to induce a prominent SAC response under these conditions unless BUB1 was depleted by an siRNA. These data are consistent with the notion that residual expression of BUB1 could be found in our previously published "Ex3" clone (Zhang *et al*, 2019). Overall, we conclude that although BUB1 is not an absolutely essential component of the SAC (cells can induce a prominent SAC response upon nocodazole treatment and even in response to noscapine treatment, the cells took ~100 min on average to exit mitosis), BUB1 does deliver a critical contribution to the SAC when the signal that activates the SAC is limited. Importantly, under physiological conditions it is more likely that a SAC response is elicited from just a few kinetochores rather than from the majority of kinetochores at the same time. Thus, loss of BUB1 is likely to affect chromosome stability, as it results in a failure to properly delay anaphase in cells where single misattachments persist for long periods of time.

This study highlights the caveats of CRISPR/Cas9 technology in generating gene knockouts. Importantly, the only way to guarantee that there is no residual expression of alternative variants is to delete the full gene locus. Altogether, by deleting the full BUB1 locus in HAP1 cells, we showed that BUB1 makes a non-essential, but important contribution to the SAC under conditions where

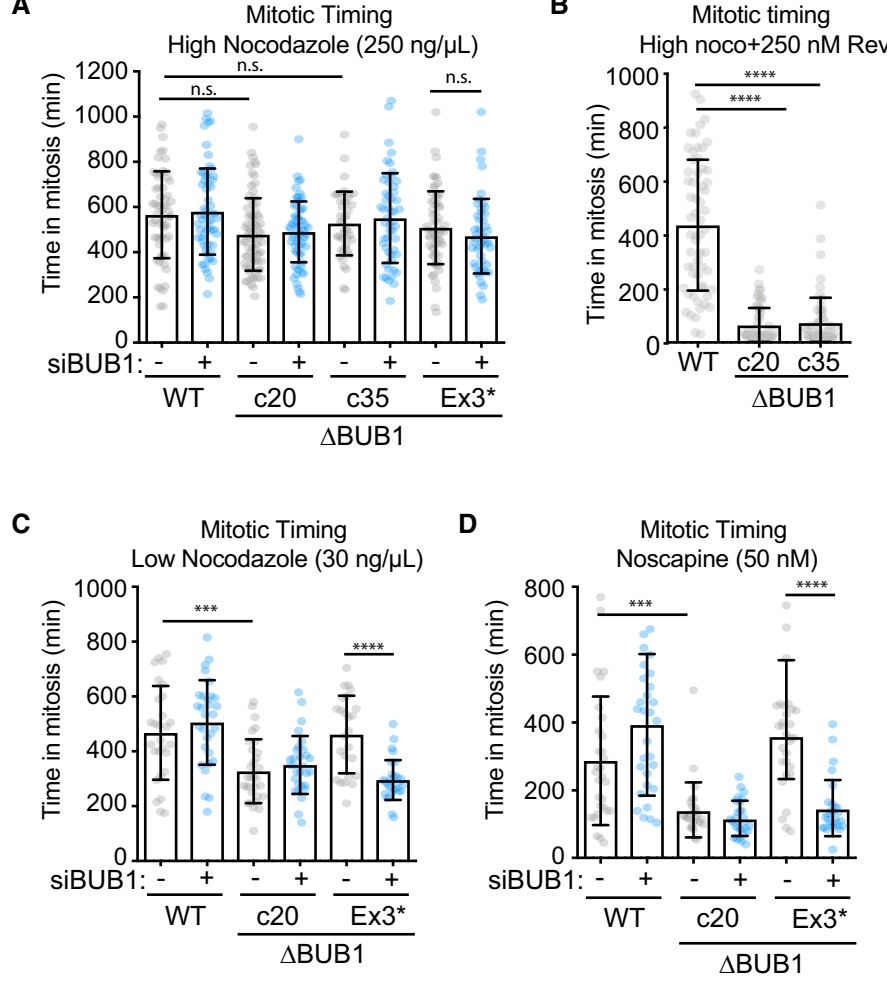

\* Raaijmakers *et al.* 2018

**Figure 2. ∆BUB1 cells display a graded SAC defect.**

(A) Average time in mitosis of cells treated with a high dose of nocodazole. DNA was visualized by SiR-DNA. Bars represent time from nuclear envelope breakdown until chromosome decondensation or death in mitosis for N = 45–80 cells per condition from two independent experiments. Dots represent data from individual cells. Gray dots represent mock-transfected cells, while the blue dots indicate cells transfected with siBUB1, 48 h prior to the start of the movie. Clones 20 and 35 are described in this study, and the clone "Ex3" is from our previous publication (Raaijmakers *et al*, 2018). Error bars represent SD. (B) Average time in mitosis of cells treated with a high dose of nocodazole + 250 nM reversine. N = 60 cells per condition from two independent experiments. Error bars represent SD. (C) Same as (A), but here cells were treated with a low dose of nocodazole instead. N = 25–31 cells per condition from a single experiment. Error bars represent SD. (D) Same as (A), but here cells were treated with noscapine instead. N = 30 cells per condition from a single experiment. Error bars represent SD.

Data information: See Appendix Supplementary Methods for experimental details.

the SAC is only partially activated. Importantly, besides HAP1 cells, our attempts to generate BUB1 knockout cells in other cell lines (HCT116, RPE-1, 293T, U2OS) have been unsuccessful. Although generating knockouts in these cell lines can be technically more challenging due to the presence of multiple alleles, this result could indicate that the loss of BUB1 is not well tolerated in all cell types. This possibly reflects a difference in the relative contribution of BUB1 to the SAC between cell lines or a difference in the tolerance toward mild chromosome segregation errors. Thus, although the most reliable way to fully resolve the exact contribution of BUB1 to the SAC in different genetic backgrounds would be to stably eradicate the full gene, this is only feasible if this is a tolerated condition.

**Expanded View** for this article is available online.

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
