## [Review Process File · The EMBO Journal]

Killing a zombie: a full deletion of the BUB1 gene in HAP1 cells

J.A. Raaijmakers and R.H. Medema.

Review timeline:

Submission date:	7 th May 2019
Editorial Decision:	23 rd July 2019
Revision received:	18 th September 2019
Accepted:	1 st October 2019

Editor: Hartmut Vodermaier

Transaction Report:

1st Editorial Decision

23rd July 2019

Thank you for sending in your correspondence and response, related to our earlier publication of the article by Zhang et al., "Efficient mitotic checkpoint signaling depends on integrated activities of Bub1 and the RZZ complex". Both letters have now been assessed by the two reviewers of the original study, whose comments are copied below for your information. As you will see, the referees appreciate the achievement of full BUB1 knock-out in HAP1 cells and several of the arguments made in both pieces, but also both feel that the points made still leave the overall issue of BUB1's general importance for spindle checkpoint signaling to some extent equivocal. In the absence of direct experimental tests, the referees consider explanations invoking RNAi off-target effects or nocodazole concentration dependencies rather speculative, and also do not find the suggestion of cell line idiosyncrasies or possible adaptations completely satisfying.

Following discussions with our Chief Editor, we conclude that for the benefit of the field, we could still offer to publish this exchange, albeit in more concise and factual format and with less speculation. In this respect, we would advocate several specific text modifications for each piece, as detailed below. Please note that like most other journals, we adhere to the policy of allowing only a single round of correspondence exchanges, irrespective of whether they are able to definitively resolve the argument, and we cannot accommodate later additions. We realize that a more definitive resolution may require further experimentation (including, but not restricted to, testing RNAi constructs and nocodazole dosing, as suggested by the referees), but we would not expect you to invest further time and resources on this for the purpose of this correspondence. That said, should you mutually agree on conducting such further work and -if necessary- adjusting the correspondence according to its outcome, we would certainly be happy to wait and accommodate this; only we will not consider adding data or follow-up exchanges once the correspondence has been accepted.

Please let me know which solution you prefer in this situation. Should you both opt for publication of the exchange in the current form, please modify your respective pieces as detailed below, and return them to us at your earliest convenience.

REFeree REPORTS

Referee #1:

The communication between the Medema and Nilsson labs attempts to resolve the role of BUB1 in the spindle checkpoint. Following various conflicting studies on this, the Medema and Millar groups reported on generation of BUB1 knockout cell lines that were viable and capable of mounting a considerable checkpoint response. This was subsequently questioned by papers from the Jallepalli and Nilsson labs, reporting that the BUB1 knockout lines did not have complete BUB1 loss-of-function. This was argued on the basis of quantitative proteomics and by the fact that BUB1 RNAi was able to fully kill the checkpoint response in the lines presumed to be BUB1 knock-out. The Medema lab now reports on the generation of two new haploid HAP1 cells lines in which in effect the entire BUB1 gene is deleted. These cells are by all account BUB1 null cells, and have an intact, albeit weakened, checkpoint response, suggesting BUB1 is not essential for the checkpoint, at least in HAP1 cells. They hypothesise that the effects of RNAi on the knock-out cells shown by Nilsson could have been off-target, and/or that the contribution of BUB1 to the SAC is cell type specific. Nilsson counters by proposing that either the concentration of nocodazole may explain the different responses and/or that the knock-out lines have adapted to the loss of BUB1 function.

Although the effort of the Medema lab to resolve this issue by generating full gene deletions is laudable, and the field should be informed on the existence of such lines and their phenotypes, in the end the exchange between the two labs is not very helpful to the community, in my opinion. Yes, genetic BUB1 deletions can be generated, but why don't the authors then take the necessary next step and perform the RNAi experiment also? Their argument that this may have been off-target is weak, as the checkpoint phenotype of the KO/RNAi combo was rescued by expression of a RNAi-resistant BUB1 in the Nilsson EMBOJ study. The Medema lab could have easily tried this in their old and new lines, and assess whether, according to their hypothesis, the RNAi effect was likely off-target. Their other argument that the contribution of BUB1 to the checkpoint may be cell type specific also is not very helpful, as this is a common way to explain away discrepancies without actually offering an explanation. The authors' statement that they have not been able to generate BUB1 knockouts in diploid lines, if anything, would argue that HAP1 lines are the odd-one-out, for whatever reason.

The response by the Nilsson lab is confusing. Besides the fact that it seems hastily written and at times somewhat defensive, it confirms that the new HAP1 lines are true nulls, but suggests that adaptation has occurred without assessing if that is so. A long list of proteomics data means little without an experiment to examine if BUB1 knockout lines live by virtue of adaptation. Regardless, the fact that these lines can be made and are viable long-term, suggest that BUB1 is not essential in HAP1 cells. The explanation of Nilsson that the difference between their experiments and Medema's may be due to difference in nocodazole concentrations used to assess the checkpoint response is a little odd: is he suggesting that his study assessed checkpoint strength in the wrong way by using a nocodazole dose that cannot assess true checkpoint loss of function?

In summary, the new data from Medema and the exchange between the labs shows that BUB1 can be deleted in HAP1 cells, but leaves us with no resolutions as to the role of BUB1 in the spindle checkpoint. My feeling is this thing won't be resolved until someone manages to generate a diploid cell system in which BUB1 protein can be removed acutely and penetrantly.

Referee #2:

In this correspondence, Raaijmakers and Medema respond to a recent article from the Nilsson group on the exact contribution of human Bub1 on the spindle assembly checkpoint activity. A first study from Raaijmakers and Medema had shown that a CRISPR knock-out of Bub1 in the haploid human HAP1 cell line that deleted a short gene fragment of Bub1 only impaired the spindle checkpoint in sensitized conditions, i.e. when the Mps1 checkpoint kinase was partially inhibited. A more recent study from the Nilsson laboratory showed, however, by mass-spectroscopy that this cell line still expressed low levels of Bub1, most likely due to alternative splicing mechanisms, identified by Rodriguez-Rodriguez et al., 2018. The Nilsson group further reported that depletion of the residual Bub1 with an siRNA led to stronger, but not full impairment of the checkpoint activity. In this present correspondence Raaijmakers and Medema generate a new, full Bub1 CRISPR KO in HAP1

cells, as they remove nearly all the exon of Bub1, and show that such a mutant cell line still has a reasonably strong checkpoint activity. Moreover, they report that such a full KO was not possible in other cell lines, such as HeLa or RPE1 cells, which could point to difference between cell lines. In a response to this correspondence, the Nilsson group first independently confirms that the newly generated does not express any residual Bub1, but disagrees with the interpretation of the Raaijmakers and Medema study that their reported Bub1 siRNA phenotype is likely due to an off-target effect

Overall, when reading the both the initial correspondence from Raaijmakers and Medema and the response from the Nilsson group, I feel that both should be published, as they significantly advance the checkpoint field and address an issue that could be poisoning the field for a while. There is, however, one controversy that would remain: whether the siRNA used by the Nilsson laboratory has a weak off-target effect on the SAC response in HAP1 cells. An off-target effect could explain the difference in results obtained in both systems, but the Nilsson laboratory used a complementation assay to rule out this possibility; alternatively it could be a difference in the concentration of nocodazole used to elicit a SAC response. Finally due to their haploid status HAP1 cells might be have a SAC response that is less sensitive to Bub1 presence, as acknowledged by both groups.

One set of experiments that has the potential to resolve this last controversy, would be to use the Bub1 siRNA of the Nilsson laboratory on the new HAP1 Bub1 cell line of Raaijmakers and Medema and to test for SAC activity by live cell imaging. If it has an off-target effect it should change the SAC response even in a cell line lacking Bub1; if the authors see no effect of that Bub1 siRNA it would suggest that the differences seen in both studies are due to the distinct nature of the used perturbations in combination with a partially functioning SAC. While the two texts could be published as such, I feel that such a last set of experiments, which could be carried out by both groups if they agree, would resolve this question in a constructive manner. While I don't know whether "revision" experiments can be part of such a correspondence, it would avoid the field a last lingering controversy.

1st Revision - authors' response

18th September 2019

In response to your previous email and the suggestions made by the reviewers, we decided to address the majority of the concerns raised as the role of BUB1 has been an ongoing deliberation and any clarification would only help the field forward. This involved that addition of several experiments; hence it took some time to complete.

The additional experiments have made several important contributions:

1. We have now repeated the experiments where we study the role of BUB1 in the SAC in low nocodazole concentrations, similar to the experimental conditions used in the Nilsson lab. Consistently with their results, we did observe a significant defect in the SAC in the full BUB1 deletion mutants (Figure 2C). Because of this result, we performed an additional experiment where we treated cells with Nocapine, resulting in minor misalignments (Figure 2D). In this condition, the SAC defect was even more prominent, suggesting that BUB1 does deliver a critical contribution to the SAC when the signal that activates the SAC is limited. We have now added this data to the revised manuscript.
2. With the aforementioned experiments, we added our previously generated Δ BUB1 clone (Raaijmakers et al. 2018 Cell Rep) as the Nilsson lab did find residual expression of BUB1 in this clone. Again, consistent with their findings, we found that this clone does not display a SAC defect under conditions where the SAC is partially activated, in contrast to the full BUB1 deletion mutants. Also consistent with their data, we find that BUB1 depletion by siRNA does induce a SAC defect in this clone, suggesting that the residual 2% of BUB1 is sufficient to promote at least a bit of SAC activity.

Taken together, our data still supports a model in which BUB1 contributes to the functionality of the SAC but its contribution is not absolutely essential under conditions of full SAC activation. However, our new data suggests that BUB1 does become important under conditions of partial SAC

activation. Our data is now fully in line with the data of the Nilsson lab and we have communicated our results with them already. We hope that you agree that this correspondence makes an important contribution to the field and you will consider this revised manuscript for publication.